# Coherent diffractive imaging of single helium nanodroplets with a high harmonic generation source

Daniela Rupp[1], Nils Monserud[2], Bruno Langbehn[1], Mario Sauppe[1], Julian Zimmermann [1], Yevheniy Ovcharenko[1,3], Thomas Möller[1], Fabio Frassetto[4], Luca Poletto[4], Andrea Trabattoni[4,5], Francesca Calegari[5,6], Mauro Nisoli[6,7], Katharina Sander[8], Christian Peltz[8], Marc J. Vrakking[2], Thomas Fennel [2,8] & Arnaud Rouzée[2]

Coherent diffractive imaging of individual free nanoparticles has opened routes for the in situ analysis of their transient structural, optical, and electronic properties. So far, single-shot single-particle diffraction was assumed to be feasible only at extreme ultraviolet and X-ray free-electron lasers, restricting this research field to large-scale facilities. Here we demonstrate single-shot imaging of isolated helium nanodroplets using extreme ultraviolet pulses from a femtosecond-laser-driven high harmonic source. We obtain bright wide-angle scattering patterns, that allow us to uniquely identify hitherto unresolved prolate shapes of superfluid helium droplets. Our results mark the advent of single-shot gas-phase nanoscopy with lab-based short-wavelength pulses and pave the way to ultrafast coherent diffractive imaging with phase-controlled multicolor fields and attosecond pulses.

[1] Institut für Optik und Atomare Physik, Technische Universität Berlin, Hardenbergstraße 36, 10623 Berlin, Germany. [2] Max-Born-Institut für Nichtlineare Optik und Kurzzeitspektroskopie, Max-Born-Straße 2A, 12489 Berlin, Germany. [3] European XFEL GmbH, Holzkoppel 4, 22869 Schenefeld, Hamburg, Germany. [4] CNR, Istituto di Fotonica e Nanotecnologie Padova, Via Trasea 7, 35131 Padova, Italy. [5] Center for Free-Electron Laser Science, DESY, Notkestr. 85, 22607 Hamburg, Germany. [6] CNR, Istituto di Fotonica e Nanotecnologie Milano, Piazza L. da Vinci 32, 20133 Milano, Italy. [7] Department of Physics, Politecnico di Milano, Piazza L. da Vinci 32, 20133 Milano, Italy. [8] Institut für Physik, Universität Rostock, Albert-Einstein-Straße 23, 18059 Rostock, Germany. Correspondence and requests for materials should be addressed to D.R. (email: daniela.rupp@physik.tu-berlin.de) or to T.F. (email: thomas.fennel@uni-rostock.de) or to A.Rée. (email: arnaud.rouzee@mbi-berlin.de)

Single-shot coherent diffractive imaging (CDI) with intense short-wavelength pulses became possible just recently with the advent of extreme ultraviolet (XUV) and X-ray free-electron lasers (FEL)[1]. This lensless imaging method has revolutionized the structural characterization of nanoscale samples, including biological specimens[2], aerosols[3], atomic clusters[4–6], and nanocrystals[7]. By capturing high-quality diffraction patterns from a single nanoparticle in free flight using a single laser pulse, the sample morphology can be determined in situ and free from spurious interactions due to deposition on a substrate. For sufficiently regular structures the wide-angle scattering information even reveals the full three-dimensional particle shape and orientation[6–8], as multiple projections of the same particle are encoded in a single diffraction image[6]. These unique capabilities enable the investigation of metastable or transient states that exist only in the gas phase. Pioneering FEL experiments have explored this frontier and demonstrated CDI of quantum vortices in helium droplets[5], ultrafast nanoplasma formation[9], and explosion of laser-heated clusters[10]. Using XUV and soft X-ray high harmonic generation (HHG) sources for single-shot nanoparticle CDI holds the promise to combine the nanoscale structural imaging capabilities of CDI with the exquisite temporal, spectral, and phase control inherent in the use of optical lasers, including the fascinating prospect of CDI with isolated attosecond pulses.

The brightness of HHG sources is typically orders of magnitude lower than that of an FEL[11], but over the years, a number of them have been scaled up in order to achieve high intensities and/or high average power[12–17], allowing CDI of fixed targets[18–23]. Experiments on ion-beam-edged nanostructures in membranes achieved impressive resolution on the order of 20 nm after multiple exposures[22, 23] and even yielded the overall shape from a single-shot diffraction image in favorable cases[20]. Here we report the feasibility of lab-based 3D characterization of unsupported nanoparticles and demonstrate single-shot HHG-CDI on individual free helium nanodroplets.

## Results

### Single-shot scattering with a high harmonic source
In our experiment (Fig. 1), a high power Ti:sapphire laser amplifier was used to generate 35 fs laser pulses at 792 nm wavelength with up to 33 mJ pulse energy. About 12 mJ were loosely focused ($f = 5$ m) into a xenon-filled gas cell[24], producing ≈2 μJ of XUV radiation, i.e., close to $10^{12}$ photons per pulse. This corresponds to ~1% of the pulse energy that can currently be achieved at the XUV free-electron laser FERMI[25]. The CDI application requires high fluence, and thus tight focusing. Typical back-reflection multilayer mirrors, however, conflict with the use of straylight apertures and

the detection of scattered light at small scattering angles. Therefore, CDI-compatible grazing-incidence microfocusing optics with an overall transmission of 10%, based on a coma-correcting system of toroidal mirrors[26], were used to focus the multicolor XUV beam to a small spot (beam-waist $\omega_0 = 10$ μm), achieving a power density of $3 \times 10^{12}$ W cm$^{-2}$ (pulse averaged). The XUV spectrum (11th to 17th harmonic, see Fig. 2d) was obtained prior to the CDI measurements with a grating spectrometer. A jet of helium droplets with diameters of several hundreds of nanometers crossed the focus of the XUV beam. The droplets were generated using a cryogenically cooled pulsed valve maintained at a temperature between 4.9 and 5.7 K, operating at low repetition rates of 3–10 Hz. The diffracted radiation was measured shot-to-shot with a wide-area MCP (micro-channel plate)-based scattering detector (see "Methods" section).

### Analysis of the multicolor diffraction patterns
Within $3 \times 10^5$ single-shot measurements, 2300 bright patterns with distinct structures were obtained and another 12,700 recorded images contained weak, unstructured scattering signal. A selection of exemplary diffraction patterns is displayed in Fig. 3. To analyze the diffraction patterns, the multiple spectral components of the HHG pulses have to be taken into account. The simultaneous use of several different wavelengths complicates the analysis of the size and shape of the particles, but it is also a fundamental precondition for generating attosecond pulse trains and isolated attosecond pulses[27]. Therefore, all future approaches toward attosecond diffractive imaging will require a multicolor analysis. We developed a multidimensional Simplex optimization[28] on the basis of multicolor Mie scattering calculations[29, 30] to analyze the diffraction images of spherical helium droplets and to demonstrate that the fits can give access to the optical properties of the droplets.

The majority of bright diffraction patterns (≈76%) shows ring structures that can be assigned to spherical droplets (cf. Fig. 2a). The diffracted field from dielectric spheres illuminated with a single wavelength can be described by the Mie solution and yields scattering patterns showing concentric rings[29, 30]. The ring separation scales with the wavelength and with the inverse particle size. The overall scattering strength and the detailed shape of the ring structure further depend on the material's complex refractive index (see "Methods" section). In our case, the XUV pulses contain spectral contributions from four harmonics (Fig. 2d). Therefore, the scattering pattern is a superposition of the corresponding single-wavelength scattering intensities, and displays a characteristic beating pattern due to the wavelength-dependent ring spacings of the individual spectral contributions to the image (*black curve* in Fig. 2b, c). The observed patterns are fitted via a multidimensional Mie-based optimization with the

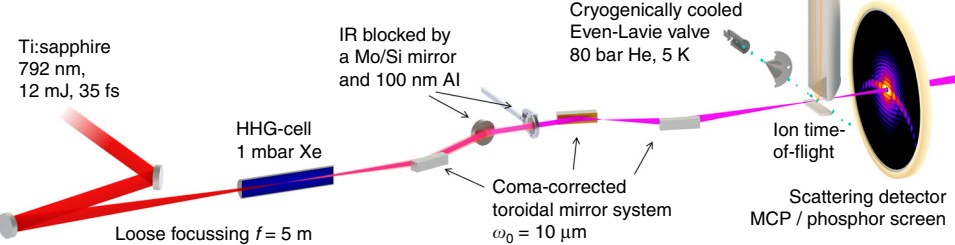

**Fig. 1** Scheme of the experimental setup. A Ti:sapphire laser with 792 nm central wavelength and 35 fs pulse duration is used for the generation of high harmonics. Up to 12 mJ are loosely focused into a xenon-filled cell, where the extreme ultraviolet (*XUV*) pulses are produced. The copropagating near-infrared (*NIR*) beam is removed via a Mo/Si mirror and a thin aluminum filter. The beam is focused to a small spot ($\omega_0 = 10$ μm) using a coma-correcting system of three gold-coated toroidal mirrors[26]. A pulsed jet of helium nanodroplets ($\overline{R} \approx 400$ nm) is overlapped with the XUV focus. The overlap is optimized by monitoring the formation of He$^+$ ions using an ion time-of-flight spectrometer. The scattering signal is amplified by a pulsed MCP and converted to optical photons on a phosphor screen. The single-shot diffraction images are captured with an out-of-vacuum camera (not depicted)

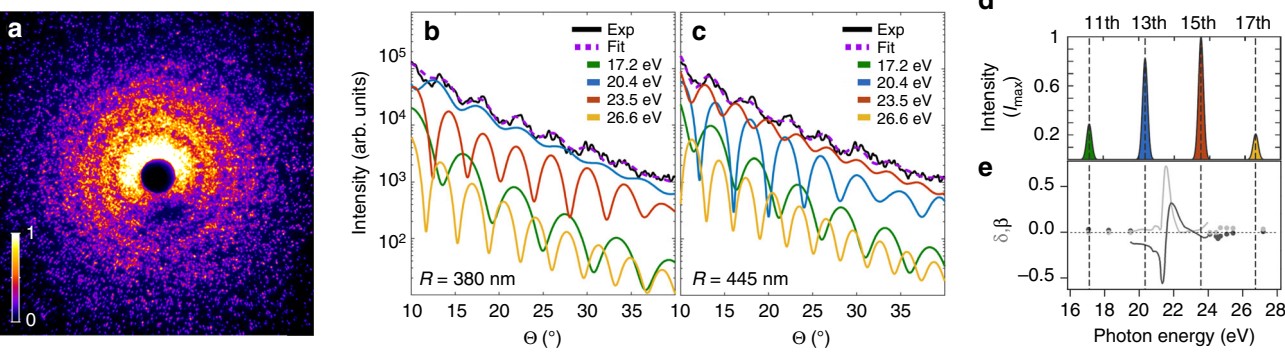

**Fig. 2** Multicolor analysis of the diffraction images. **a** Measured bright scattering image (center part of the detector, intensity in arbitrary units) from a spherical droplet with a pronounced concentric ring pattern. **b, c** Multicolor Mie fits (*dashed purple*) of the extracted radial intensity profile (*solid black*) from **a** as obtained via a simplex optimization (see "Methods" section) of the individual harmonic contributions to the profiles (*color-coded in green, blue, red, and yellow*). The results illustrate that two qualitatively different solutions yield comparably small residuals. The two solutions indicate that either the 13th harmonic **b** or the 15th harmonic **c** dominates. The resulting refractive indices of these and all other fits are given in Supplementary Fig. 4. **d** Measured average extreme ultraviolet spectrum of the high harmonic radiation. **e** Sketch of the energy-dependent refractive indices of bulk liquid helium in the vicinity of the helium 1s–2p transition, assembled from bulk liquid helium measurements[31, 32] (*solid lines, color-coded in light-gray and dark-gray*) and tabulated values (*scatter*, NIST database, http://physics.nist.gov/PhysRefData/FFast/html/form.html)

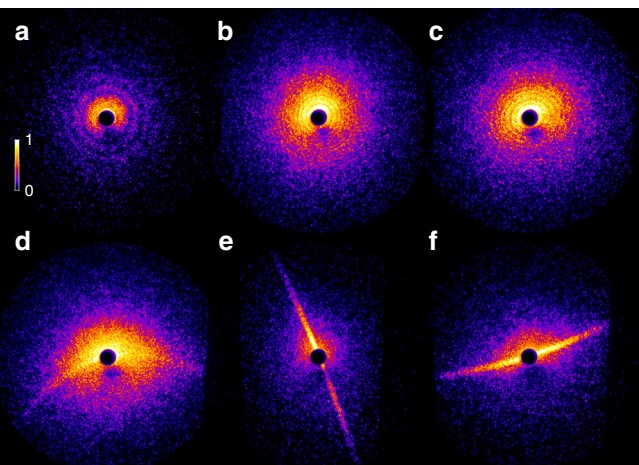

**Fig. 3** Three characteristic types of diffraction images from helium nanodroplets. The majority of images contains concentric ring patterns **a**, **b** that are assigned to spherical droplets. Elliptical ring structures as in **c** or pronounced streak patterns as in **d–f** reflect deformed helium droplets. The measured dataset comprises 1762 ring-type, 421 elliptical, and 68 streak-type images. In most cases, the latter exhibit a clear bending of the streaks (55 out of 68 images), e.g., as in **d** and **f**. For details, see Supplementary Note 3. False color images, color bar applies to all patterns, intensity in arbitrary units

particle size, the refractive indices at the wavelengths of the contributing harmonics, the relative intensities of the harmonics, and a scaling factor for the total intensity of the XUV pulse as input parameters (see "Methods" section). While the optical properties of bulk liquid helium have been measured and calculated close to the 1s–2p transition of helium[31, 32] (see Fig. 2e), the dielectric function of the nanodroplets is completely unknown and expected to vary substantially with droplet size[33]. In fact, we find that fits using the bulk literature values for the refractive indices at the corresponding harmonic wavelengths cannot reproduce the observed diffraction patterns (see also Supplementary Note 1).

Successful fits can be achieved by using the refractive indices at the wavelengths of the dominant 13th and 15th harmonics as optimization parameters in addition to the particle size. In this

procedure, the relative intensities of the contributing harmonics are set to measured average values and the refractive indices at the wavelengths of the 11th and 17th harmonics are fixed, as they lie far away from the large helium resonances (cf. Fig. 2e). The optimization was successfully carried out for 18 very bright scattering patterns with clear beating structures up to large scattering angles. The majority of these fits indicate a dominant contribution of the 13th harmonic, as exemplified in Fig. 2b. However, for all patterns, a second solution with dominant signal from the 15th harmonic is found by the algorithm with comparably low residuals of the fit (cf. Fig. 2c and Supplementary Note 2). As the similarly intense 13th and 15th harmonics lie close to each other (only 3.1 eV energetic distance), the fitting algorithm can vary the refractive indices freely and exchange their roles in the fit, while adjusting the cluster size accordingly (compare Fig. 2b, c). We note that while the residuals of the fits are slightly smaller for the solution with dominant 13th harmonic, the refractive indices for the solution with dominant 15th harmonic lie closer to the literature values of bulk helium. However, we cannot fully exclude one of the two solutions. In order to resolve such ambiguity in the optimization, future systematic studies are required with only one of the strong harmonics being near-resonance and/or with substantially better signal to noise ratio. By using higher energy and/or lower wavelength lasers to drive the HHG process, it is anticipated that the photon flux of individual harmonics can be further increased by at least one order of magnitude[17], while the energetic distance between the harmonics becomes larger. However, the present analysis supports that the multicolor fit procedure can be used for a new metrology of optical parameters and constitutes a basis toward future multicolor imaging approaches. In the subsequent scattering simulations, the average values of the refractive indices from the solutions with dominant 13th harmonic are used (see "Methods" section).

**Identification of prolate droplet shapes.** Besides concentric ring patterns (Fig. 3a, b) and an abundance of about 20% of elliptical patterns from ellipsoidal droplets (cf. Fig. 3c), about 3% of all bright images exhibit pronounced streak structures as exemplified in Fig. 3d–f and Fig. 4a. The abundances of the three main types of patterns, i.e., rings, elliptical and streak patterns, are similar to what has been reported in previous hard X-ray measurements at

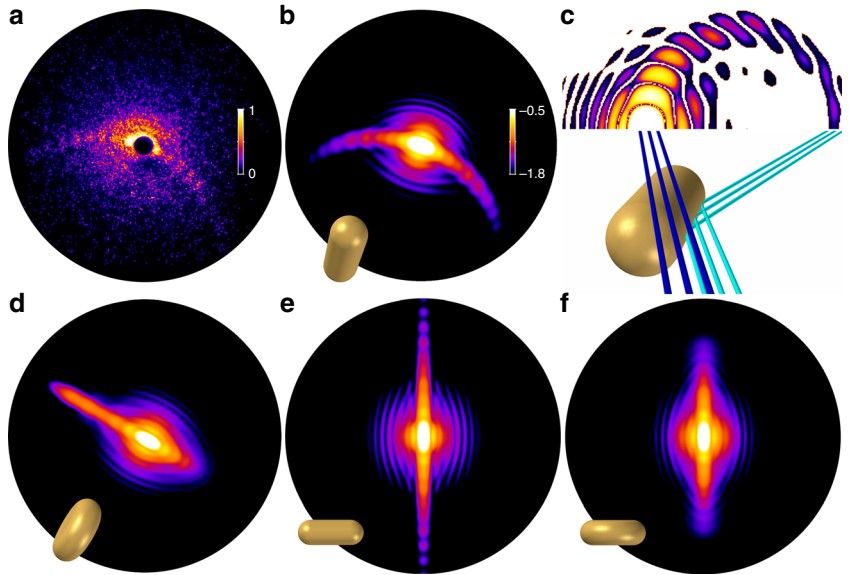

**Fig. 4** Unique identification of prolate pill-shaped structures. **a** Measured image (intensity in arbitrary units) and **b** matching simulation result of the wide-angle diffraction of a pill-shaped prolate droplet (logarithmic intensity in arbitrary units, color bar applies also to **d**–**f**). The structure shape and orientation are visualized in *yellow*. The optical axis of the extreme ultraviolet beam is directed into the image plane, the tilt angle between the symmetry axis of the particle and the optical axis is 35°; the semi-minor axes $a = b = 370$ nm and the semi-major axis $c = 950$ nm; for optical parameters see "Methods" section. **c** Illustration of the origin of bent streaks occurring when a tilted rod-type structure diffracts the light. The constructive interference is analogous to the specular reflection at the surface of a macroscopic rod. Two particular bundles of constructively interfering rays are explicitly sketched, please note that the different ray colors do not refer to wavelengths, but are applied to facilitate distinction. **d** Simulated wide-angle diffraction image of a wheel-shaped oblate particle (semi-major-/-minor axes as in **b**, tilt angle of 80° between the symmetry axis and the optical axis, which is directed into the image plane). If the oblate particle's symmetry axis is neither oriented along the optical axis nor perpendicular to it, the diffraction patterns exhibit straight streaks to only one side. **e**, **f** Comparison of simulated wide-angle diffraction images of a prolate **e** and an oblate structure **f** aligned to the scattering plane, i.e., at 90° tilt angle between the symmetry axis and the optical axis, other parameters as in **b**, **d**. Though the 2D projections are similar and the 2D outlines identical, the intensity distributions of the straight streaks are clearly different and decay much faster for wheel-type than for pill-type shapes

LCLS[5]. However, whereas straight streak patterns were observed in the X-ray results, we find in the majority of our wide-angle scattering patterns a pronounced crescent-shaped bending of the streaks (statistics see Fig. 3 and Supplementary Note 3). In the X-ray experiments, the reconstruction of the corresponding droplet shapes via iterative phase retrieval revealed 2D projections of the helium droplets with extreme aspect ratios[5]. These were assigned to extremely flattened, wheel-like oblate shapes. The deformation was attributed to a high angular momentum, which can be transferred to the droplets by cavitation and rip-off from the liquid phase during the formation process[5, 34], while vibrational excitations are assumed to decay very quickly[35]. Whereas classical viscid rotating droplets undergo a deformation from oblate to prolate two-lobed shapes with the rotation axis perpendicular to the long axis of the droplet[36, 37], this transition has been suggested to be hindered in helium nanodroplets by the appearance of vortex arrays that deform the droplets and stabilize extreme oblate shapes[38]. Very recently, the occurence of such classically unstable oblate helium droplets was further supported by statistical arguments, while an indication of rare prolate structures was also found[39]. However, a unique discrimination of prolate and oblate shapes based on the 2D projections accessible with small-angle X-ray diffraction is difficult[39]. Tomographic information, on the other hand, is contained in XUV wide-angle scattering and can be exploited to retrieve the three-dimensional particle shape and orientation provided the particle morphology is sufficiently regular[6, 7]. In our measurement, the presence of tomographic information prominently manifests in the bending of the streak patterns.

In order to retrieve the shapes underlying the experimentally observed scattering patterns, three-dimensional multicolor scattering simulations were performed[40] (see "Methods" section). The two-sided, bent streak features can only be reproduced, considering prolate droplets as shown in Fig. 4b, matching the experimental pattern of Fig. 4a. Crescent-shaped streaks arise from prolate structures that are tilted out of the scattering plane (i.e., the plane normal to the laser propagation axis). The wide-angle interference pattern can be intuitively understood in analogy to the reflection of a laser from a macroscopic rod. As shown in Fig. 4c, bundles of rays diffracted by the cylindrical part of the surface gather the same path length and interfere constructively. In contrast, a tilted wheel-shaped particle with the same aspect ratio as exemplified in Fig. 4d cannot explain the bending of the streaks. Instead, in the wide-angle scattering regime, the simulations indicate that such particles would generate a one-sided straight streak, a phenomenon that was not observed in our experiment. Although less obvious, also the analysis of observed straight streak patterns indicates that they can only be explained by prolate structures, as the observed streak signal is visible until the edge of the detector (cf. for example Fig. 3e). In figure 4, panels e and f show a comparison of diffraction patterns for pill- and wheel-shaped structures, that are aligned to the scattering plane. The tomographic nature of wide-angle scattering reveals that the streaks decay much faster toward larger scattering angles for a wheel than for a pill-shaped particle. We would like to note that the absence of wheel-shaped droplets in our experiment may stem from the different droplet generation scheme, using a pulsed valve with a long nozzle in this work, where the transition to the superfluid state might be delayed compared to the short CW flow nozzle used by Gomez et al.[5]. However, our unambiguous observation of prolate droplets, which are known to occur for classical liquids[36, 37],

will contribute to the discussion on the stability of spinning superfluid droplets. Their existence may provide a fascinating case for future experiments, as it should be possible to clarify if a prolate droplet shows macroscopic-shape rotation, which is not expected for a superfluid droplet[38].

## Discussion

We have shown the feasibility of single-shot single-particle CDI using intense XUV pulses from a HHG source. Bright diffraction patterns of spherical helium droplets have been obtained and matched with simulations using optimized refractive indices. The observed crescent-shaped streak patterns could be uniquely assigned to prolate droplets. The results further suggest several future prospects connected to the HHG-specific properties. Laser-based HHG provides a high accessibility compared to FEL facilities in a wavelength regime suitable for 3D shape characterization of non-reproducible gas-phase nano-objects, particularly if experiments with single harmonics can be realized, which is anticipated using UV or deep UV driver lasers for HHG[17, 41]. This will facilitate fundamental investigations of structure formation, such as tracing ice nucleation[42], with important implications for atmospheric physics and aerosol science. Moreover, unprecedented experiments beyond structural determination are possible, such as multicolor tomography and resonant-pump–resonant-probe CDI, which exploit the time-resolution and phase control achievable in HHG-based experiments for diffractive imaging of quantum coherent dynamics. The spatiotemporal characterization of ultrafast electron dynamics has been driving attosecond science[27, 43] from the beginning and will perhaps be the most exciting prospect of HHG-CDI. Considering the advancing capability of generating intense isolated attosecond pulses[15] and the possibility of stroboscopic illumination using attosecond pulse trains[44], the vision of diffractive imaging of attosecond electron dynamics in isolated nanostructures has come in reach.

## Methods

**Femtosecond laser system and generation of XUV harmonics**. The experiments are performed using a commercially available cryo-cooled Ti:sapphire laser amplifier (KMLabs Red Wyvern) delivering pulses at a central wavelength of 792 nm with 33 mJ pulse energy and 35 fs pulse duration at 1 kHz repetition rate. A fraction of 30%, of the output energy (typically 10–12 mJ), are taken for the generation of high harmonics. To this aim, a broadband spherical mirror with a focal distance of 5 m is used to focus the near-infrared (NIR) pulses into a 100 mm long aluminum gas cell statically filled with ≈1.3 mbar of xenon (loose focusing geometry). The position of the gas cell, the gas pressure and the NIR pulse energy are adjusted to optimize the HHG flux. An output energy of ~2 µJ (measured with a calibrated photodiode) is achieved in this geometry, corresponding to a conversion efficiency of $1.6 \times 10^{-4}$ and an average power of 2 mW. To the best of our knowledge, this is the highest average power obtained by means of HHG. The harmonic beam consists of the 11th (72 nm), 13th (61 nm), 15th (53 nm), and 17th (47 nm) harmonics (see Fig. 2d) as measured by dispersing the XUV beam with a grating spectrometer. The XUV pulse duration was characterized in previous experiments using THz electron streaking technique to be roughly 20 fs.

**Microfocusing setup and IR filter**. A high throughput XUV beamline (transmission ≈10%) consisting of three gold-coated, grazing incidence (10°) toroidal mirrors and a flat Mo/Si mirror is used to tightly focus the XUV beam in the interaction region. Positions, radii and distances between the toroidal mirrors were optimized by ray tracing in order to achieve a high demagnification factor of 25 for the XUV beam while keeping the coma-aberrations low[26, 45]. The first 40 mm × 10 mm toroidal mirror with radii 57.6 m × 1.735 m is placed 5 m away from the gas cell in order to collimate the XUV beam. The collimated beam is then reflected by a flat Mo/Si mirror that partially absorbs the co-propagating NIR laser pulse used for HHG. The remaining NIR laser beam (≈1 mJ) is filtered out by a 100 nm thin aluminum filter. We note that the reduction of the NIR by the Mo/Si mirror is required to avoid damaging the aluminum foil. A coma-corrected system composed of two toroidal mirrors facing each other is then used to demagnify and tightly focus the XUV beam into the experimental chamber. The first of the latter two toroidal mirrors (radii 2650 mm × 79 mm) has a focal length of 230 mm. Subsequently, the last toroidal mirror (radii 3620 mm × 109.2 mm) is placed 680 mm away from the focus to relay image the focus of the first toroidal mirror into

the experimental chamber at a distance of 585 mm. This geometry allows a demagnification factor of 25 with respect to the initial XUV spot size at the generation point. Considering a 0.5 mrad × 0.5 mrad diverging XUV beam with a 175 µm spotsize (FWHM) at the source point, we expect to achieve a minimum spot size at the focus of 7 µm (FWHM). In our experiment, the size of the XUV beam was characterized by monitoring the fluorescence of a Cs/YAG screen placed at the focus using a CCD camera. We measured a 9 µm × 10 µm spot size (FWHM) with a 10 µm beam waist, in close agreement with the expected value, leading to an intensity of $I_f = 3 \times 10^{12}$ W cm$^{-2}$.

**Helium droplet generation**. The helium nanodroplets are generated with a pulsed Even-Lavie valve[46] that is cooled with a Sumitomo closed-cycle cryostat down to 4.9–5.7 K. The minimum temperature depends on the repetition rate and the opening duration of the valve (varied between 3 and 10 Hz and 18 to 27 µs, respectively), which influence the heat load of the valve. High-purity ${}^4$He (99.9999%) at a pressure of 80 bars is expanded into a differentially pumped UHV chamber through a 100 µm trumpet-shaped nozzle located at 450 mm distance to the interaction region. The droplet pulse is guided into the interaction chamber through a conical skimmer with 1 mm diameter, which reduces the uncondensed gas in the interaction chamber.

**Scattering experiment**. A large-area scattering detector (⌀: 75 mm) with a center hole (⌀: 3 mm) is placed 37 mm behind the XUV focus, corresponding to a maximum spatial frequency of 0.09 nm$^{-1}$ for the dominant wavelength of 53 nm. The detector consists of a Chevron-type MCP for signal amplification and a phosphor screen for conversion to optical light[4]. The MCP is used in pulsed operation to suppress background signal from charged particles. Further, the CDI-compatible focusing geometry described above allows for the use of two straylight apertures before the focus to minimize photonic background signal on the detector. The 8° tilt of the MCP channels results in an area with decreased response observable at the lower right side of the detector hole (cf. Figs. 2a, 3 and 4a)[47]. The scattering patterns are recorded on a shot-to-shot basis using an out-of-vacuum CMOS camera. An ion time-of-flight spectrometer is used for establishing and optimizing the spatial overlap of the XUV pulses and the helium nanodroplets and the timing of the droplet jet[48]. Within $3 \times 10^5$ single-shot measurements 2300 bright patterns with distinct structures were obtained. Further 12,700 recorded images contained weak, unstructured scattering signal. These statistics indicate that the experiment is performed in the single-particle limit as the probability to have two droplets in the focus at the same time is <2‰.

**Data analysis and scattering simulations**. For comparison with theory, the measured diffraction patterns were transformed to the scattered intensity that would be recorded on a spherical detector. In addition, the measured data must be corrected for the nonlinear detection efficiency of the MCP[6, 9]. Previous work has shown that the saturation effect can be described by an exponential efficiency function[6] such that the detected signal intensity, $I_{det}$, is connected to the true experimental intensity, $I_{exp}$, via $I_{det} = I_{exp}^\alpha$. The nonlinearity exponent $\alpha = 0.5$ has been found by matching the angular decay of the envelope of scattering profiles from spherical droplets to the universal $q^{-4}$ decay behavior predicted by Porod's law[49, 50]. The center position was independently determined for every pattern to correct for slight variations resulting from wavefront tilts at the position of the droplet[51]. Radial intensity profiles were extracted by angular averaging over the upper half of the detector to avoid any influence from the area on the MCP, where the detection efficiency is decreased since the incoming photons impinge on the MCP parallel to the MCP channels[47] (cf. Methods, Scattering experiment). In order to fit the measured patterns for spherical particles, we employed a multi-dimensional Simplex optimization[28] on the basis of multicolor Mie scattering calculations[29, 30]. The intensity pattern in a calculated multicolor image contains four single-frequency-components (11th to 17th harmonic), each weighted with the intensity of the respective harmonic order. As fitting parameters, the particle size and the photon energy-dependent refractive indices ($n = 1 - \delta + i \cdot \beta$ with $\delta$ being the deviation of the real part of the refractive index from unity and $\beta$, the imaginary part, which corresponds to the absorption) at the 13th and 15th harmonic were varied. For helium, the 11th and 17th harmonics are far away from resonances[31, 32], so that tabulated literature values of the refractive indices could be fixed within the fit procedure (17.2 eV: $n = 0.97 + i0.0$; 26.6 eV: $n = 0.9964 + i0.041$) (values taken from the NIST data base http://physics.nist.gov/PhysRefData/FFast/html/form.html). A set of 18 measured patterns with clear rings up to maximum scattering angle were fitted using a hybrid Monte–Carlo Simplex optimization algorithm for a large ensemble of trajectories (see Supplementary Figs. 5 and 6). Each fitting trajectory was initialized with random start parameters in a reasonable range for the corresponding optimization parameters ($R = 300$ nm to $R = 600$ nm, $\delta = -0.3$ to $\delta = 0.2$, $\beta = 0$ to $\beta = 0.07$), and subsequently improved via Simplex optimization. The scattering patterns for nonspherical shapes were calculated in the discrete-dipole-approximation as implemented in ref. 40 by the superposition of four single-color calculations and using the average optical parameters of the 13th and 15th harmonic determined in our study (solution with dominant 13th harmonic, 20.4 eV: $n = 0.9252 + i0.0178$; 23.5 eV: $n = 1.2688 + i0.0417$). The custom computer codes used are available on request from the authors.

**Data availability**. All the data used in this study are available on request from the corresponding authors.

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

## Acknowledgements

The authors kindly acknowledge Bernd Schütte's excellent work for the initial development of the HHG beamline. D.R. thanks Andrey Vilesov, Christoph Bostedt, Bernd von Issendorff and Joachim Ullrich for helpful and enlightening discussions. Excellent support has been provided by the TUB-IOAP workshop. This project has received funding from DFG (Grants No. MO 719/13–1 and /14–1), from BMBF (Grant No. 05K13KT2), and from the European Union's Horizon 2020 research and innovation programme under the Marie Skłodowska–Curie grant agreement No. 641789. Further, T.F. acknowledges computational resources provided by the North-German Super-computing Alliance (HLRN) and financial support from the Deutsche For-schungsgemeinschaft via SFB652/3, a Heisenberg Fellowship (grant No.: FE 1120/4–1) and from BMBF (grant No.: 05K16HRB). F.C. and M.N. acknowledge funding from ERC grants STARLIGHT (grant No. 637756), and ELYCHE (grant No. 227355).

## Author contributions

D.R. and Y.O. performed the feasibility studies in advance of the experiment. F.F., L.P., A.T., F.C., and M.N. developed the microfocusing optics setup and implemented it together with N.M. and A.R. The helium jet was set up by B.L., D.R. and M.S. set up the CDI detection system, and J.Z. developed the data acquisition system. N.M. and A.R. operated the HHG source, and D.R., M.S., B.L., N.M. and A.R. assembled and carried out the experiment. K.S., C.P., and T.F. developed and performed the scattering simulations. D.R., N.M., B.L., J.Z., K.S., C.P., and T.F. analyzed the data with input from all authors. The manuscript was discussed and written with input from all authors.

## Additional information

**Competing interests:** The authors declare no competing financial interests.

