## [Peer Review File · Nature Communications]

Reviewers' comments:

Reviewer #1 (Remarks to the Author):

The field of CDI has been growing exponentially with time. This manuscript extends the previous work (their reference 7) using FLASH to an HHG source and the fascinating problem of superfluid helium droplets. The concept to use 'long' wavelengths was emphasized in the work of the Fennel group that is somewhat missing in this paper. In addition there is the use of several harmonics and fitting to extract optical properties. While the conceptually simple it is almost certainly better to use a single wave-length from perhaps a FEL and take advantage of the 3d information from 'long' wavelength CDI. Of course there is the $\lambda/2$ resolution limit which long wavelength experiments suffer. However they provide 3d information on a single shot that is inaccessible with shorter wavelengths.

The paper alludes to attosecond imaging using pulses in the cutoff region of the HHG spectrum but no where do they 'calculate' that there are enough photons for single shot experiments nor do they justify the need.

The main points of the paper as I see them are: (1) one can do CDI with an HHG source, the authors comment on more photons but that's a wait and see and (2) the advantages of using long wavelengths, not explicitly discussed that have been addressed in reference 7.

To make the paper acceptable I believe these concepts have to be clarified in a revised manuscript and quantitative comparisons with a tunable, near transform limited FEL source, FERMI for example, which works in this energy range need to be included.

With revisions I think this paper represents something of sufficiently broad interest to be published.

Reviewer #2 (Remarks to the Author):

This manuscript describes how a multi-color "table-top" HHG XUV laser can be used to image the shapes of sub-micron superfluid helium droplets. The authors tried to fit the collected single-shot diffraction patterns to multi-color Mie scattering profiles by varying two out of four complex refractive indices of Helium within the relevant energy range — the description in the manuscript suggests that these two indices were not well known from literature. Further, the authors argue that the shape of some of the superfluid helium droplets were prolate rather than oblate, which was hitherto unobserved.

While the manuscript shows a fascinating application of multi-color HHG CDI, some of its key conclusions lack convincingly supportive evidence or description. Specifically, the controls and considerations listed below should be addressed before the authors' conclusions can be better assessed.

Regarding fitting the refractive indices:

- The manuscript shows only the maximum pulse intensities of the 4 harmonics but do not show the pulse-to-pulse energy fluctuations between these four harmonics. This fluctuation appears important as admitted by the authors "... scatter of the fitting results remains large, which is mainly attributed to shot-to-shot fluctuations in the XUV spectrum..". How much do the fluctuations between harmonics affect the Simplex fitting? Are the ratios of their energies constrained during the fits? One can imagine that reasonable fits could be achieved by fixing the refractive indices but letting the pulse energies vary during the fitting as well.
- How mono-disperse were the injected He droplets? This is especially relevant here given that the droplets' responses to the 15th harmonic appears to be very sensitive to the droplet sizes [ref 28].

A cautious attempt at a droplet-size distribution or an estimate of its variance can help quantify the effects of size-dependence.

- The fitted delta and beta parameters vary noticeably from pattern to pattern. Could this be related to the size fluctuations of He droplets? It seems odd that the fluctuations in the 13th harmonic is mostly in beta, while that of the 15th is mostly in delta? Further, deviations from the 13th are strictly in +delta, while those of 15th are strictly in -delta. These curious patterns, if genuine, appear statistically significant and bear a serious quantitative discussion.
- How were the angular averages performed on the diffraction intensities (e.g. Fig 2a,c) prior to fitting? How did the authors account for the seemingly dimmer intensities in the lower right portion of all their diffraction patterns (also shown in Fig 3a)?
- Have the authors considered that their pulses might suffer phase tilt fluctuations from shot to shot [Sensing the wavefront of x-ray free-electron lasers using aerosol spheres. Optics Express, 21(10), 12385–12394.]? The patterns presented in the paper are superficially symptomatic of such tilt fluctuations. Such phase tilts could cause apparent loss of speckle contrast when angularly averaging over a mis-identified center. And a loss of speckle contrast could be "fixed" by varying the scattering contributions by various harmonics. To get around this issue, one could consider performing 2D Mie fits instead of 1D fits, where one could easily adjust the center of each measured pattern using centro-symmetry at lower spatial frequencies prior to the fitting.

Regarding the droplet shape analysis:

- The prolate shape analysis is intriguing. Unfortunately, the justification for the existence of such shapes points to a paper still in preparation. The non-specialist reader would appreciate a motivation about why rare prolate shapes are expected to occur at all.
- Can the authors show that these prolate droplets cannot arise from dynamic shape fluctuations from droplets during injection? If possible, it would help if these shapes can be shown to disappear at temperatures far above the superfluid transition.

The conclusions drawn by this manuscript appear hastily drawn from insufficient analyses or poor explanations, or both. In both of these regards, the current manuscript requires strong revisions.

Reviewer #3 (Remarks to the Author):

This work reports imaging of superfluid helium droplets using the high harmonics of a femtosecond laser. The work is innovative, and the analysis is solid. The writing, on the other hand, leaves much room to be desired for: some sentences start with abbreviations, some even start with a number, and some statements are incomprehensible. There are a few minor content issues as well.

1. Pg. 5, discussion on the fitted refractive indices: The oscillation for the 13th harmonic seems to be much weaker than the rest. Is it due to resonance? Which state in resonance?
2. Among the three types of images, what type of diffraction accounts for the straight lines? A simulation of the rod shaped droplet lined up with the diffraction plane should be provided for reference, either in the main text or supplementary material. A statistics of the images for in-plane and out-of-plane rods and spherical droplets should be provided.

Grammatical issues:

1. Pg. 2, line 33, I am not sure if the imaging method in this article can be considered "tomographic" since no cross-sectional information is obtained.
2. Pg. 2, line 38, unclear sentence "would allow to combine..."
3. Pg. 3, sentence start with an abbreviation.
4. Pg. 4, line 64, "A dataset of 2300 ...", should state the total images taken, the hit rate, and the total number of bright images.
5. Pg. 5, line 96, "using higher energy and/or lower wavelength drivers", what do the authors mean by "drivers"? Lasers?

6. Pg. 5, line 100, "these three main types of patterns": state the three main types explicitly.
7. Pg. 8, line 150, starting a sentence with "30%"?
8. Pg. 9, line 184, "230 mm away from the mirror": does it mean that the focal length is 230 mm?
9. Pg. 10, line 204, "statistics indicate": what statistics?
10. Pg. 11, line 208 – 209, why do the authors assume the detected intensity to scale in that formula? Why use $\alpha = 0.5$?
11. Fig. 1, a problem with the labeling for "IR-blocker, MoSi mirror..."
12. In general, I find the figures are difficult to visualize on my computer, and impossible to print, particularly Fig. 2. Too many not quite relevant pictures are placed together without much logic.

(comments from Referee 1 are printed in **violet** and our response in **black**)

Referee 1: “The field of CDI has been growing exponentially with time. This manuscript extends the previous work (their reference 7) using FLASH to an HHG source and the fascinating problem of superfluid helium droplets.”

We thank the reviewer for highlighting the fascinating science emerging from single-particle CDI of superfluid helium droplets with HHG sources. We respond to the specific comments in detail below.

(1a) “The concept to use 'long' wavelengths was emphasized in the work of the Fennel group that is somewhat missing in this paper.”

The comment seems to refer to old Ref. 6 (Barke et al., Nat. Comm. 6, 6187 (2015)). In this publication, the collaboration around Ingo Barke, including Thomas Fennel and several other authors of our manuscript, has developed a conceptually new method to retrieve the morphology of individual clusters based on “long” wavelength wide-angle scattering. We have now applied this technique for the first time in a lab-based experiment. The current first application of the HHG wide-angle technique to the very interesting system of superfluid Helium droplets immediately led to a fascinating new result, i.e. the observation of prolate droplet

shapes, contradicting previous work from the X-ray free-electron laser LCLS (old Ref. 5, Gomez et al., Science 345, 906-909 (2014)). We thus feel that our current work represents a substantial advance compared to the previous paper [Barke2015] and clearly cites/discusses the importance of the previous work on the wide angle technology. We modified the introduction to emphasize the relation to the previous work more clearly.

changes to the manuscript:

To discuss previous results more prominently, we change the corresponding sentence in the introduction to *“For sufficiently regular structures the wide-angle scattering information even reveals the full three-dimensional particle shape and orientation [Barke2015, Reines2009], as multiple projections of the same particle are encoded in a single diffraction image [Barke2015].”*

(1b) *“In addition there is the use of several harmonics and fitting to extract optical properties. While the conceptually simple it is almost certainly better to use a single wave-length from perhaps an FEL and take advantage of the 3d information from 'long' wavelength CDI. Of course there is the $\lambda/2$ resolution limit which long wavelength experiments suffer. However they provide 3d information on a single shot that is inaccessible with shorter wavelengths.”*

On the one hand, concerning the exclusive shape characterization, we fully agree that a multicolor pulse is a complication in the sense of a blurring/overlay effect while single harmonic or monochromatic FEL pulses produce “clean” patterns that are much easier to interpret. On the other hand, it is likely that the feasibility of lab-based 3D imaging experiments will lead to breakthroughs in many scientific areas, even with pulses containing multiple harmonics. Moreover, it is certainly fair to assume that single harmonic sources will be available very soon for such experiments. In fact, for future HHG-based shape characterization studies, we have already started development towards using 400nm-based HHG, or even 266nm as a driver wavelength (for work from other groups see new Ref. [Cirmi2012] and Ref. [Popmintchev2015]) that can be tuned to increase the energy spacing between harmonics that can then be filtered out. The demonstrated feasibility of HHG imaging in free flight with multiple harmonics from an 800nm driver laser is just a technical detail and by no means limits the science addressed with our experiment.

On the other hand, the multicolor nature is not only a disadvantage. Previous work has shown that the multicolor combination can even enhance the spatial resolution ([Chen2009], old Ref. 21). In the current work we show that it is likely to enable a new metrology for the single-shot characterization of optical properties of free particles. In future studies, this novel concept will allow to determine the size-dependent refractive indices of helium droplets, which are unknown (see also [Joppien1993], old Ref. 28). At the current stage, the presence of 2 strong harmonics near the helium resonance complicates the analysis as it leaves some ambiguity, i.e. two possible sets of optical parameters that fit the results reasonably well (see discussion below

(2a)). This shows that there is substantial room for improvement from this first experiment, however, the current results do not put fundamental restrictions on the technology.

Most importantly, the feasibility of single-particle HHG-CDI with multicolor-pulses constitutes a milestone towards time-resolved diffraction experiments with attosecond resolution, which is not possible with free-electron lasers yet. For any future use of both attosecond pulses and/or attosecond pulse trains in CDI experiments one must inevitably deal with the effect of broad spectra in the analysis. Therefore we strongly believe that it will pay off to scrutinize multicolor pulses in single particle diffractive imaging. We revised the manuscript to reflect these arguments more clearly.

[Cirimi2012] G. Cirimi et al., J. Phys. B 45 205601 (2012).

[Popmintchev2015] D. Popmintchev et al., Science 350, 1225–1231 (2015)

[Chen2009] B. Chen et al., Phys. Rev. A 79, 023809 (2009)

[Joppien1993] M. Joppien et al., Phys. Rev. Lett. 71, 2654–2657 (1993)

changes to the manuscript:

- We included a statement on single harmonic sources to the conclusion paragraph: “, *particularly if experiments with single harmonics can be realized, which is anticipated using UV or deep UV driver lasers for HHG [Cirimi2012, Popmintchev2015].*” The citation [Cirimi2012] was added.
- Concerning the need of multicolor analysis tools we added “*To analyze the diffraction patterns, the multicolor components of the HHG pulses have to be taken into account. The simultaneous use of several different wavelengths complicates the analysis of the size and shape of the particles, but it is also a fundamental precondition for generating attosecond pulse trains and isolated attosecond pulses [Krausz2009]. Therefore all future approaches towards attosecond diffractive imaging will require a multicolor analysis. We developed a multidimensional Simplex optimization [Lagarias1998] on the basis of multicolor Mie scattering calculations [Mie1908, Bohren1983] to analyze the diffraction images of spherical helium droplets and to demonstrate that the fits can give access to the optical properties of the droplets.*”

(1c) “The paper alludes to attosecond imaging using pulses in the cutoff region of the HHG spectrum but no where do they 'calculate' that there enough photons for single shot experiments nor do they justify the need.”

We would like to reply that we have convinced ourselves of (i) the principle feasibility to image single particles with HHG based attosecond pulses in the near future and (ii) the capability to reach the necessary characteristics in our own lab. The current experiments were performed using typically 10-12 mJ laser pulses from a commercial Ti:Sapphire system (32 mJ @ 792 nm

central wavelength with 35 fs duration). The remaining 20 mJ can be used to pump a high energy TOPAS (light conversion HE) that allows to generate mid-infrared pulses with a pulse energy above 4 mJ and a tunable central wavelength between 1100 nm and 2400 nm. Recently, Midorikawa and co-workers [Takahashi2013] have demonstrated gigawatt-scale isolated attosecond pulses (1.3 μJ , 500 as pulse duration) by combining a two-color field (800 nm: 9 mJ+1300 nm: 2.5 mJ) for the high harmonic generation process. Clearly, these parameters are well within what is currently available in our laboratory. We therefore expect that by adding a mid-infrared pulse we will be able to generate isolated attosecond pulses with a pulse energy above 1 μJ with our experimental setup, i.e. with a similar pulse energy used in the investigation presented in our manuscript.

In the near future, we will also have the opportunity at the Max-Born Institute to perform attosecond pump-probe experiments using an Optical Parametric Chirped Pulse Amplifier (OPCPA) that, at the same central wavelength as the afore-mentioned Ti:Sa laser system, will produce ~ 30 mJ laser pulses with a pulse duration ≤ 10 fs. At the current stage of the OPCPA development, using only a part of the available pump power, 12 mJ pulse energies have already been achieved. From this it follows that the OPCPA will allow experiments, where the XUV yield that we have used in the current experiments (obtained using the commercial laser system) is a lower limit, with significant potential for further improvement on the basis of the shorter pulse duration and the higher pulse energy of the OPCPA.

Still, as suggested by the referee, it remains an open question at the moment, whether isolated attosecond pulses (IAPs) can be generated with a flux that is sufficient for meaningful coherent diffractive imaging, given that the generation of IAPs is typically accompanied by a substantial (easily 1-2 orders of magnitude) loss in XUV yield. However, the recent development of (i) the two-color laser scheme for high-order harmonic generation and (ii) OPCPA technologies have already shown the large potential to reach this goal with intense IAPs. Furthermore, experiments using attosecond pulse trains (APT) are certainly possible. In particular when using a stroboscopic illumination (i.e. using APTs where the generation yields one attosecond pulse per infrared optical cycle [Mauritsson2008]), attosecond time-resolved imaging of IR-induced plasma waves should be possible. We therefore consider that our current study represents a first step towards attosecond coherent diffractive imaging experiments. The spatiotemporal characterization of attosecond electron dynamics has been seen as a key motivator and major goal of attosecond science by its pioneers [Niikura2007, Krausz2009]. Here attosecond CDI constitutes a promising approach, in particular for imaging collective electron motion and strong field plasma formation dynamics in finite systems such as clusters and nanoparticles.

[Takahashi2013] E. J. Takahashi, P. Lan, O. D. Mücke, Y. Nabekawa and K. Midorikawa, "Attosecond nonlinear optics using gigawatt-scale isolated attosecond pulses" *Nature communication* 4, 2691, (2013).

[Mauritsson2008] J. Mauritsson, P. Johnsson, E. Mansten, M. Swoboda, T. Ruchon, A. L'Huillier and K. J. Schafer, "Coherent electron scattering captured by an attosecond stroboscope" *Phys. Rev. Lett.*, 100, 073003, (2008).

[Niikura2007] Niikura, H. and Corkum, P. "Attosecond and angstrom science". *Advances In Atomic, Molecular, and Optical Physics* 54, 511 – 548 (2007).

[Krausz2009] Krausz, F. and Ivanov, M. "Attosecond physics". *Rev. Mod. Phys.* 81, 163–234 (2009).

changes to the manuscript:

- To justify the need of and the interest in attosecond CDI, and to make the considerations of feasibility more comprehensible we changed the last part of the outlook paragraph to:
"The spatiotemporal characterization of ultrafast electron dynamics has been driving attosecond science [Niikura2007, Krausz2009] from the beginning and will perhaps be the most exciting prospect of HHG-CDI. Considering the advancing capability of generating intense isolated attosecond pulses [Takahashi2013] and the possibility of stroboscopic illumination using attosecond pulse trains [Mauritsson2008], the vision of diffractive imaging of attosecond electron dynamics in isolated nanostructures has come in reach."

(1d) "The main points of the paper as I see them are: (1) one can do CDI with an HHG source, the authors comment on more photons but that's a wait and see and (2) the advantages of using long wavelengths, not explicitly discussed that have been addressed in reference 7."

We thank the referee for acknowledging that the demonstration of CDI with HHG is a key result. Regarding the advantages of using long wavelengths, we believe that the comment refers again to [Barke2015], our previous work at FLASH, i.e. old Ref. 6. In this case we have addressed this concern in point (1a). We would like to stress again, that applying the wide-angle technique using HHG pulses to the very interesting system of superfluid Helium droplets led to the fascinating observation of prolate droplet shapes, that will add an important new perspective to the vivid current discussion of superfluid droplet shape (please see also discussion below (2f)).

The second way to interpret her/his comment, "not explicitly discussed that have been addressed in reference 7", is that the referee sees the need to emphasize the work by the Miao group ([Xu2014], old ref 7) more to give appropriate credit. We feel that we firmly cited the technological basis described in previous work, including the important work by Xu et al. However, we decided to offer the following change to the manuscript.

changes to the manuscript:

- To give more credit to the paper mentioned by the referee we included it more prominently in the introduction in the sentence *"This lensless imaging method has revolutionized the structural characterization of nanoscale samples including biological specimens [Seibert2011], aerosols [Loh2012], atomic clusters [ClustersAtFLASH, Gomez2014, Barke2015], and nanocrystals [Xu2014]."*

(1e) “To make the paper acceptable I believe these concepts have to be clarified in a revised manuscript and quantitative comparisons with a tunable, near transform limited FEL source, FERMI for example, which works in this energy range need to be included.”

Regarding the clarification of the concepts, we have revised the text in order to:

- better address the advantages, disadvantages, and opportunities of multicolor pulses (cf. the above response and changes under **1a** and **1b**)
- estimate technical feasibility and better motivate attosecond CDI (cf. the above response and changes under **1c**)
- stronger emphasize the work mentioned by the referee (cf. the above response and changes under **1d**)

The remaining point mentioned by the referee is the “quantitative comparisons with a tunable, near transform limited FEL source, FERMI for example, which works in this energy range”, which is provided below.

For our experiments, pulses with about 2 μJ pulses were generated; this corresponds to approximately 6×10^{11} XUV photons per pulse (without beamline transmission). This compares to pulse energies of up to 300 μJ that can be achieved at the FERMI free-electron laser [Allaria2012, FermiUserWebpage] for a single wavelength in comparably long pulses (several tens of fs). Thus one can say that the used HHG pulses deliver two orders of magnitude lower pulse energy than comparable FEL sources. However, this disadvantage of HHG sources is to a significant extent offset by the fact that table-top HHG sources can easily run at higher repetition rate, permit virtually unrestricted access, already provide better time-resolution than comparable FEL sources, and will in the near term even allow entering the attosecond domain. For this reason, we believe that CDI experiments at FELs and using table-top HHG sources are to be regarded as being complementary, each with specific strengths and weaknesses, and together providing a unique CDI toolbox.

[Allaria2012] Allaria, E. et al., “Highly coherent and stable pulses from the FERMI seeded free-electron laser in the extreme ultraviolet” Nature photonics. 6, 699–704 (2012)

[FermiUserWebpage] <https://www.elettra.trieste.it/lightsources/fermi/fermi-machine>

changes to the manuscript:

- Following the referees suggestion, we added the FERMI citations to the manuscript, stating “About 12 mJ were loosely focused ($f = 5 \text{ m}$) into a xenon-filled gas cell, producing approximately 2 μJ of XUV radiation, i.e. close to 10^{12} photons per pulse. This corresponds to approximately 1% of the pulse energy that can currently be achieved at the XUV free-electron laser FERMI [Allaria2012, FermiUserWebpage].”

(1f) “With revisions I think this paper represents something of sufficiently broad interest to be published.”

We feel that we could clarify all referee concerns. The changes have greatly increased the accessibility of the text such that we are confident that it should now be ready for publication.

----- **Response to referee 2** -----

(comments from Referee 2 are printed in **red** and our response in **black**)

Referee 2: “This manuscript describes how a multi-color “table-top” HHG XUV laser can be used to image the shapes of sub-micron superfluid helium droplets. The authors tried to fit the collected single-shot diffraction patterns to multi-color Mie scattering profiles by varying two out of four complex refractive indices of Helium within the relevant energy range — the description in the manuscript suggests that these two indices were not well known from literature. Further, the authors argue that the shape of some of the superfluid helium droplets were prolate rather than oblate, which was hitherto unobserved.

While the manuscript shows a fascinating application of multi-color HHG CDI, some of its key conclusions lack convincingly supportive evidence or description. Specifically, the controls and considerations listed below should be addressed before the authors’ conclusions can be better assessed.”

We thank the reviewer for the supportive evaluation and the challenge expressed by the referee to significantly improve the presentation and conclusions of our manuscript. We have responded to this challenge and, after analyzing substantially more data, have changed the presentation and discussion of our fitting results for the refractive indices. In short, the expanded analysis has substantiated the fact that two solutions with similar residuals can be found, impeding a unique determination of optical properties in the current experiment. The reason is explained in detail below (2a). On this basis we clarify that the multicolor fits are currently mainly a technical requirement to analyze the data. We underline that multicolor tools will become a necessity for attosecond CDI experiments and point out the current difficulties and sketch pathways to resolve them.

We feel that the two key results of our study, i.e. (1) the demonstration HHG-based CDI and (2) the identification of so far unresolved helium nanodroplet shapes, in conjunction with the various future implications, ranging from structural analysis of nondepositable specimen to CDI of attosecond electron dynamics, justify the publication of our revised manuscript in Nature Communication.

(2a) “Regarding fitting the refractive indices: The manuscript shows only the maximum pulse intensities of the 4 harmonics but do not show the pulse-to-pulse energy fluctuations between these four harmonics. This fluctuation appears important as admitted by the authors “.. scatter of the fitting results remains large, which is mainly attributed to shot-to-shot fluctuations in the XUV spectrum..”. How much do the fluctuations between harmonics affect the Simplex fitting? Are the ratios of their energies constrained during the fits? One can imagine that reasonable fits could be achieved by fixing the refractive indices but letting the pulse energies vary during the fitting as well.”

The analysis of the diffraction patterns requires the account of the multicolor nature of the experiment. Importantly, the analysis shows that the observed scattering patterns cannot be explained with the literature values for the refractive indices of bulk liquid helium. Instead, they can be described with refractive indices optimized in the fit procedure. In our initial submission of the manuscript we showed the best fits but did not perform a deeper, systematic analysis of the resulting values (i.e. their dependence on the droplet size). Although we now analyzed substantially more patterns for the revision of the manuscript, we still cannot provide such systematic analysis due to a remaining ambiguity of the resulting refractive indices which we would like to explain in detail first. After that we will come back to the discussion of the fluctuation effect as suggested by the referee.

In our extended analysis we were able to fit 18 diffraction patterns of spherical droplets. The analysis confirms our previous conclusion that the patterns cannot be explained with literature values for the refractive indices. This also holds when allowing the fit algorithm to freely vary the intensity ratios, as suggested by the referee. The thus achieved best fits with fixed refractive

Figure R1 a)-c) Multicolor-Mie fits with fixed refractive indices (literature values for bulk liquid helium). Intensity ratios are used as fitting parameters. Best fits are shown in purple, harmonic contributions are color coded as indicated in the legend. The fit quality is seriously reduced compared to the best fits with variable refractive indices and harmonic intensity ratios fixed to their average measured values (cf. Fig. R4, revised Fig. 3 of the manuscript, and Figs. S5 and S6 in the supplemental materials).

indices (Fig. R1 shows 3 examples) have a substantially higher residual error (a factor >3 worse, cf. Fig. R5). Furthermore, the intensity ratios associated with these best fits have a very unlikely distribution, as displayed in Figure R2. For example the most intense contributions in these fits stem from the 17th harmonic that has been measured to be the weakest. Again, we thus conclude that the refractive index literature values cannot explain the data.

For the 18 patterns fitted with intensity ratios fixed to the measured ones, the majority of the best fits lead to refractive indices that lie around the results presented in the initial submission. In these cases, the 13th harmonic gives the dominant contribution to the diffraction image, as for example shown in old Fig. 2. However, for all analyzed patterns, a systematic second solution is found by the algorithm (such an example was already shown in the supplement of the initial submission). In a few cases this second solution, where the dominant contribution systematically comes from the 15th harmonic, even yields the best fit. An example of these two “good solutions” found for the same diffraction pattern is displayed in Fig. R3.

Figure R2 Intensity ratios for the best fits shown in Fig. R1. Compared to the measured average ratios presented in Fig 3e of the main manuscript they show a completely different behavior which we consider very unlikely.

Figure R3: Best fit and second best fit for one pattern, a) with a dominant contribution from the 13th harmonic and b) with an only slightly worse residual error, exhibiting an exchanged role of 13th and 15th harmonic.

The refractive indices of 13th and 15th harmonic resulting from the best and second best fits are displayed in Fig. R4. The errors of the best and second best fits (blue=13th harmonic dominant, red=15th harmonic dominant) are plotted in Fig. R5 together with the much larger errors from the fits with refractive indices fixed to the literature values and variable harmonic intensities

The main reason for the existence of two solutions leading to comparably good fitting results lies in the proximity of the similarly intense 13th and 15th harmonics (only 3.1 eV energy separation). The fitting algorithm can vary the refractive indices freely and therefore can “exchange the roles” of the 13th and 15th harmonic, while adjusting the cluster size accordingly. This can be well observed in Fig. R3.

Figure R4: a) beta vs. delta at the 13th (blue) and 15th (red) harmonic wavelength for all best fits where the 13th harmonic gives the dominant contribution. b) beta vs. delta for 15th harmonic dominant. Literature values are displayed as stars, lines indicate pairs of points resulting from the same fit.

Figure R5: Errors (mean square displacement of the logarithm of the intensity profiles) of all optimized patterns. Blue: Solution with 13th harmonic dominant. Red: Solution with 15th harmonic dominant. Green: Refractive indices fixed to literature values, intensities varied freely.

(green). While the residuals of the fits for dominant contribution from 13th harmonic are systematically slightly smaller, the results from the second solution on the other hand lie closer to the literature values. However, we cannot fully exclude one of the two solutions. Therefore we show and discuss both optimization minima in the manuscript to make it more comprehensible that we cannot provide a final systematic analysis.

It is understood that one would like to use the data right away to extract the size-dependent optical parameters. This, however, is not possible yet due to the remaining ambiguity and will require an improved experiment with only one strong harmonic near the resonance and/or with substantially better signal resolution.

As indicated in our initial submission and pointed out by the referee, fluctuation effects and the sensitivity of the optimization procedure are crucial points that deserve attention. In the fits we so far used the measured mean intensity ratios of the individual harmonics. To answer the referee's question for the fluctuation effect, we explored the robustness of the fitted optical parameters against intensity fluctuations systematically.

Typical shot-to-shot fluctuation for the employed source of +/- 10% were observed in a previous experiment; in the current experiment no single shot spectra were measured. For two exemplary patterns the fits were repeated after changing the intensity contribution of one harmonic by +/- 10% (i.e. 8 fits per pattern)). The resulting refractive indices at the 13th and 15th harmonics are depicted in Figure R6 (the lines between the points indicate which values origin

from the same fit). The rather small scatter of the results shows that the resulting optical parameters are rather robust against intensity fluctuations (the best fit always ended up in the same “good solution” with the dominant 13th harmonic). This finding leads to two conclusions:

- the robustness of the fit supports the potential of the multicolor approach for providing a useful new metrology for optical parameters
- the fluctuation effect alone cannot explain the spread of the data

The large scatter of the fitted refractive indices in Fig. R4 (even when considering only one class of the possible solutions) may thus be reflecting the true size dependence of the optical properties, though the latter cannot be finally extracted yet (cf. discussion under point 2c below). However, it has to be excluded that in our experiment the intensity fluctuations and/or the deviation from the average spectrum as obtained prior to the CDI measurement were larger than expected. Therefore future experiments have to include single-shot spectral measurements.

changes to the manuscript

- In addition to exemplary fits for fixed literature refractive indices and fixed relative intensities (in the initial submission presented as supplement Fig. S1) we added Fig. R1 and R2 to the supplement, demonstrating that also for variable relative intensities the literature values of the refractive indices cannot explain the data.
- In light of the discussion we refrained from stating as a result in the abstract of our manuscript, that the diffraction patterns “provide access to the nanostructure's optical parameters”. In the revised manuscript the sentence just reads: “We obtain bright wide-angle scattering patterns, that allow us to uniquely identify hitherto unresolved prolate shapes of superfluid helium droplets.”
- We rewrote the whole section concerning the multicolor analysis. Now it reads “Therefore the scattering pattern is a superposition of the corresponding single-wavelength scattering intensities, and displays a characteristic beating pattern due to the wavelength-dependent ring spacings of the individual spectral contributions to the image (black curve in Figs. 3b/c). The observed patterns are fitted via a multidimensional Mie-based optimization with (i) the particle size, (ii) the refractive indices at the

wavelengths of the contributing harmonics, (iii) the relative intensities of the harmonics and (iv) the intensity of the XUV pulse as input parameters (see Methods). While the optical properties of bulk liquid helium have been measured and calculated close to the 1s-2p transition of helium [Surko1969, Lucas1983] (see Fig. 3e), the dielectric function of the nanodroplets is completely unknown and expected to vary substantially with droplet size [Joppien1993]. In fact, we find that fits using the bulk literature values for the refractive indices at the corresponding harmonic wavelengths cannot reproduce the observed diffraction patterns (see supplemental section 1).

Successful fits can be achieved by using the refractive indices at the wavelengths of the dominant 13th and 15th harmonics as optimization parameters in addition to the particle size. In this procedure, the relative intensities of the contributing harmonics are set to measured average values and the refractive indices at the wavelengths of the 11th and 17th harmonics are fixed, as they lie far away from the large helium resonances [Lucas1983]. The optimization was successfully carried out for 18 very bright scattering patterns with clear beating structures up to large scattering angles. The majority of these fits indicate a dominant contribution of the 13th harmonic, as exemplified in Fig. 3b. However, for all patterns a second solution with dominant signal from the 15th harmonic is found by the algorithm with comparably low residuals of the fit (cf. Fig. 3c and supplemental section 1). As the similarly intense 13th and 15th harmonics lie close to each other (only 3.1 eV energetic distance), the fitting algorithm can vary the refractive indices freely and “exchange their roles” in the fit, while adjusting the cluster size accordingly (compare Figs. 3b and c). In order to resolve the ambiguity in the optimization, future systematic studies are required with only one of the strong harmonics being near-resonance and/or with substantially better signal to noise ratio. By using higher energy and/or lower wavelength lasers to drive the HHG process, it is anticipated that the photon flux of individual harmonics can be further increased by at least one order of magnitude [Popmintchev2015] while the energetic distance between the harmonics becomes larger. However, the analysis supports that the multicolor fit procedure can be used for a new metrology of optical parameters and constitutes a basis towards future multicolor imaging approaches. In the subsequent scattering simulations, the average values from the solutions with dominant 13th harmonic are used (Fig. 3b, cf. also supplemental section 1).”

- Fig. 2 was remade, now labeled as Fig. 3, showing one spherical pattern with both solutions.
- All 2x18 fitting results, the refractive index results from both solutions (Fig. R4), and the residual errors (Fig. R5) are presented in the supplemental material.

(2b) How mono-disperse were the injected He droplets? This is especially relevant here given that the droplets' responses to the 15th harmonic appears to be very sensitive to the droplet sizes [ref 28]. A cautious attempt at a droplet-size distribution or an estimate of its variance can help quantify the effects of size-dependence.

Figure R7: a) Size distribution for all best fits where the 13th harmonic gives the dominant contribution. Average radius $\langle R \rangle = 383$ nm. b) Size distribution for 15th harmonic dominant. Average radius $\langle R \rangle = 430$ nm.

The radii obtained from all 18 best and second best fits are shown in Fig. R7 a (13th dominant) and b (15th dominant). Characterization measurements in a different experiment (Master thesis Bruno Langbehn, publication in preparation) indicate that the droplet jet yields a rather broad size distribution with at the current conditions an average size of $\langle R \rangle = 350$ nm and a FWHM of 300 nm. This is in rather good agreement with the results from our fitting procedure (Fig. R7). However it is likely that droplets with sizes outside the observed range (250 nm - 550 nm) are also present but cannot be fitted. Smaller droplets produce a diffraction pattern with low photon statistic and very large droplets have a very small fringe separation such that minima are more likely contaminated by noise fluctuations.

changes to the manuscript

- We added Fig. R7 and the related discussion to the supplemental material, section 2.

(2c) The fitted delta and beta parameters vary noticeably from pattern to pattern. Could this be related to the size fluctuations of He droplets? It seems odd that the fluctuations in the 13th harmonic is mostly in beta, while that of the 15th is mostly in delta? Further, deviations from the 13th are strictly in +delta, while those of 15th are strictly in -delta. These curious patterns, if genuine, appear statistically significant and bear a serious quantitative discussion.

We would like to acknowledge the referee's insight; there are tendencies in our current data that we would like to show to the referee and discuss, but we consider the current results too vague for a deeper discussion in the manuscript due to the above explained ambiguities in the fit results. The refractive indices obtained in the fits are plotted as a function of droplet size in Fig. R8. There are clear trends in the size dependence for a given "good solution". For example if the 13th harmonic is dominant, the real part of the refractive index for the 13th harmonic systematically increases with size (first row, 1st panel). This trend is also observed for the

Figure R8: Left: Fitted refractive indices at 13th and 15th harmonic as a function of radius for solutions with dominant 13th harmonic. Right: Fitted refractive indices at 13th and 15th harmonic vs. radius for solutions with dominant 15th harmonic.

solutions for the 15th harmonic being dominant (first row, 4th panel). Further, in all cases, the absorption (related to β) appears to decrease with increasing cluster size. We are confident that refined experiments in the near future will resolve the ambiguity and will provide access to the size-dependent optical parameters.

changes to the manuscript

- We emphasized in our manuscript that probing the size-dependent refractive indices of nanoparticles via single shot CDI will be possible in future systematic studies, whereas here the feasibility was shown. *“In order to resolve such ambiguity in the optimization, future systematic studies are required with only one of the strong harmonics being near-resonance and/or with substantially better signal to noise ratio. By using higher energy and/or lower wavelength lasers to drive the HHG process, it is anticipated that the photon flux of individual harmonics can be further increased by at least one order of magnitude [Popmintchev2015] while the energetic distance between the harmonics becomes larger. However, the present analysis supports that the multicolor fit procedure can be used for a new metrology of optical parameters and constitutes a basis towards future multicolor imaging approaches.”*

(2d) “How were the angular averages performed on the diffraction intensities (e.g. Fig 2a,c) prior to fitting? How did the authors account for the seemingly dimmer intensities in the lower right portion of all their diffraction patterns (also shown in Fig 3a)?”

Figure R9 shows the selected detector half area of the detector that was used for calculating the intensity profiles. Thus, the dimmer area of

Figure R9: Radial profiles were obtained by averaging over the upper detector half.

the detector was excluded. The change of sensitivity of the microchannel plates with respect to the angle of incidence of the incoming radiation has been reported previously (Fukuzawa et al., J. Phys. B 49, 034004 (2016), old ref. 37).

changes to the manuscript:

- We added a sentence to the methods section “Data analysis and scattering simulation”, stating that *“Radial intensity profiles were extracted by angular averaging over the upper half of the detector to avoid any influence from the area on the MCP where the detection efficiency is decreased since the incoming photons impinge on the MCP parallel to the MCP channels [Fukuzawa2016] (cf. Methods, Scattering experiment).”*

(2e) Have the authors considered that their pulses might suffer phase tilt fluctuations from shot to shot [Sensing the wavefront of x-ray free-electron lasers using aerosol spheres. Optics Express, 21(10), 12385–12394.]? The patterns presented in the paper are superficially symptomatic of such tilt fluctuations. Such phase tilts could cause apparent loss of speckle contrast when angularly averaging over a mis-identified center. And a loss of speckle contrast could be "fixed" by varying the scattering contributions by various harmonics. To get around this issue, one could consider performing 2D Mie fits instead of 1D fits, where one could easily adjust the center of each measured pattern using centro-symmetry at lower spatial frequencies prior to the fitting.

As we have performed many CDI experiments before, we are fully aware of possible center spot variations in the scattering images due to (i) wavefront tilt and (ii) interaction point fluctuations due to the actual position of the hit particle. For the analysis presented in this manuscript, the variations of the center positions have been quantified and found to be small such that an average center position was used for the fits. We have checked the robustness of the fit results with respect to the center spot variations and found no qualitative changes. However, in light of the referee comment, we have now decided to change our fit procedure and use a center that is determined for each pattern individually and repeated the fitting procedure.

changes to the manuscript:

- Fig. 2 and all figures included in the rebuttal and the supplemental section now contain the results for center-position-corrected profiles. However, it should be emphasized that this aspect has no qualitative effect on the results.
- We have added a sentence in the methods section and added the citation proposed by the referee. *“The center position was independently determined for every pattern to correct for slight variations resulting from wavefront tilts at the position of the droplet [Loh2013].”*

(2f) “Regarding the droplet shape analysis:

The prolate shape analysis is intriguing. Unfortunately, the justification for the existence of such shapes points to a paper still in preparation. The non-specialist reader would appreciate a motivation about why rare prolate shapes are expected to occur at all. Can the authors show that these prolate droplets cannot arise from dynamic shape fluctuations from droplets during injection? If possible, it would help if these shapes can be shown to disappear at temperatures far above the superfluid transition.

We would like to thank the referee for the appreciation of our shape results. We would like to clarify that we do not cite Bernardo, Vilesov and colleagues to justify the existence of prolate structures – we want to give credit to related studies on the shape of helium nanodroplets. The paper has meanwhile been published (Bernardo, C. et al., Phys. Rev. B 95, 064510 (2017)) and the citation has been updated. Our observation of the bent streak patterns is self-contained and allows a unique assignment to prolate structures. A rod-type surface is needed to cast this kind of interference pattern. We tried to emphasize this point in the paper by adding an explanatory subfigure (see also Fig. R10). There, two particular cases are explicitly sketched for a prolate droplet tilted out of the scattering plane by 20° showing bundles of rays with zero path difference that result in an interference maximum. In light blue, a bundle of rays is diffracted by the top of the cylindric part of the droplet under grazing incidence, while the dark blue rays are diffracted by the side of the cylinder. The vanishing path difference ($\Delta s=0$) results in constructive interferences. These considerations are in full analogy to light reflection on a macroscopic rod.

Figure R10: Bent streaks occur when a tilted rod-type structure diffracts the light. Two particularly obvious bundles of rays (top and side) are explicitly sketched with zero path difference which results in an interference maximum.

Concerning the question on the origin of deformed droplets, also in previous work from the Vilesov group it has been noted that in “general, a droplet may acquire a nonspherical shape due to rotational or vibrational excitation. However, our estimates show that vibrational shape oscillations should decay before the interaction point.” (Gomez et al., *Science* 345, 906-909 (2014)). In the supplemental material of Gomez et al., the decay time of oscillatory excitations has been calculated to be in the range of $1\mu\text{s}$, which is two orders of magnitude faster than the flight time of the helium droplet to the interaction region.

The shape of classical viscous droplets results from the competition between surface tension and centrifugal force. Under increasing rotation, a droplet flattens (for example the earth has an oblate shape) but from a certain rotational momentum on it becomes energetically favorable to form triaxially prolate shapes, that tend first to a pill shape and then even to dumbbell shapes (sometimes observed for tektites that fall on earth) before they are no longer stable and fission occurs (Brown, R. and Scriven, L., *Proceedings of the Royal Society of London* 371, 331–357 (1980); Baldwin, Butler, and Hill, *Scientific Reports* 5, 7660 (2015)). It is very interesting to think about and study how superfluid droplets react on rotational momentum. As there is no friction, the whole liquid droplet doesn't really circulate but the rotational momentum is contained in the vortex lattice. In a theoretical paper following the publication of Andrey Vilesov and coworkers from Ancilotto and colleagues (Ancilotto, Pi, and Barranco, *Phys. Rev. B* 91, 100503 (2015)), it has been suggested that the vortex lattice bends the droplet surface, creating oblate shapes beyond the classical limit. The authors state that „multilobe configurations present in classical viscid droplets are hindered by the appearance of vortex arrays whose regular distribution is hard to accommodate into peanutlike (or higher lobe number) shapes“ [Ancilotto2015]. We find it a fundamentally fascinating thought that for a prolate shape classically rotating around an axis perpendicular to its long axis, the rotation is clearly visible, while oblate droplets are axially symmetric and always look the same, i.e. it is not possible to decide if the whole structure is rotating. Do the prolate droplets demonstrated in our work rotate? Does superfluidity prevent the rotation of the whole structure?

It is indeed an intriguing idea to study in a controlled way the transition from superfluid to non-superfluid by increasing the temperature. However the droplets have a temperature of 0.37 K and have been observed to evade heating by fast evaporation (during expansion from 20K to 0.5K within 10^{-7}s (Toennies and Vilesov, *Angew. Chem. Int. Ed.* 43, 2622 – 2648 (2004)). Thus, the droplets probably maintain temperatures well below the transition temperature to superfluidity at 2.7 K until they are completely evaporated. However, on the basis of our results, one could speculate that it might be possible to exit the superfluid state by increasing rotational momentum. While it would be also experimentally very challenging to try to control the rotational momentum applied to the droplets, this would be a really fascinating topic for future experiments.

changes to the manuscript:

- Following the referee's suggestion we added additional information on the physics of rotating helium droplets "In the X-ray experiments, the reconstruction of the corresponding droplet shapes via iterative phase retrieval revealed 2D projections of the helium droplets with extreme aspect ratios [Gomez2014]. *These were assigned to extremely flattened, "wheel-like" oblate shapes. The deformation was attributed to a high angular momentum, which can be transferred to the droplets by cavitation and rip-off from the liquid phase during the formation process [Toennies2004, Gomez2014], while vibrational excitations are assumed to decay very quickly [SuppMatGomez2014]. Whereas classical viscid rotating droplets undergo a deformation from oblate to prolate two-lobed shapes with the rotation axis perpendicular to the long axis of the droplet [Brown1980, Baldwin2015], this transition has been suggested to be hindered in helium nanodroplets by the appearance of vortex arrays that deform the droplets and stabilize extreme oblate shapes [Ancilotto2015]."*
- To enhance the comprehensibility of our claim, to uniquely assign bent streak patterns to prolate particles, we added the information from Fig. R11 to revised Fig. 3 and state in the text "*Crescent-shaped streaks arise from prolate structures that are tilted out of the scattering plane (i.e. the plane normal to the laser propagation axis). The wide-angle interference pattern can be intuitively understood in analogy to the reflection of a laser from a macroscopic rod. As shown in Fig. 4c, bundles of rays diffracted by the cylindrical part of the surface gather the same path length and interfere constructively.*"
- To give an outlook on the significance of prolate helium nanodroplets, the paragraph is now ended by the statement "*Their existence may provide a fascinating case for future experiments, as it should be possible to clarify if a prolate droplet shows macroscopic shape rotation, which is not expected for a superfluid droplet [Ancilotto2015]."*

(2g) The conclusions drawn by this manuscript appear hastily drawn from insufficient analyses or poor explanations, or both. In both of these regards, the current manuscript requires strong revisions.

We would like to particularly thank referee 2 again for her/ his challenging but extremely helpful comments and thoughts that helped to improve our manuscript substantially. We extended our analysis and deepened and revised our discussion, hoping to now convince the referee that the conclusions drawn in our manuscript,

- (i) HHG-CDI on free nanoparticles is feasible and suggests numerous exciting future experimental possibilities,

- (ii) the multicolor analysis reveals the incompatibility of reported refractive indices to the nanodroplets data and allows in principle to retrieve the actual optical constants and
- (iii) prolate helium nanodroplets are observed which add a fascinating twist to the discussion of superfluid droplet shapes,

are well supported by our data analysis and discussion.

----- **Response to referee 3** -----

(comments from Referee 3 are printed in **green** and our response in **black**)

Referee 3: This work reports imaging of superfluid helium droplets using the high harmonics of a femtosecond laser. The work is innovative, and the analysis is solid. The writing, on the other hand, leaves much room to be desired for: some sentences start with abbreviations, some even start with a number, and some statements are incomprehensible. There are a few minor content issues as well.

We like to thank the reviewer for the supportive statements and apologize for any inconvenience regarding text and presentation. We have revised the manuscript along the lines suggested by the reviewer - corresponding changes and our response to the comments are detailed below.

(3a) Pg. 5, discussion on the fitted refractive indices: The oscillation for the 13th harmonic seems to be much weaker than the rest. Is it due to resonance? Which state in resonance?

The referee's interpretation is correct; the depth of the minima is a function of the refractive indices. In addition, while the intensity contributions of the 13th and 15th harmonic are very similar, the clear dominance of one color in the fits hints to a resonance effect.

In the photon energy range 17-27 eV the first resonance of helium occurs ($1s^2$ to $1s2p$ transition, at 21.4 eV in the atomic case). As the symmetry is broken for droplets at the surface, also the $1s^2$ to $1s2s$ transition is partially allowed, which lies at slightly lower energy. Below this transition there is no possible excitation in helium and the droplets should be completely transparent. Previous work suggests that the optical properties of helium depend on the droplet size (M. Joppien et al., Phys. Rev. Lett. 71, 2654–2657 (1993)), which is fully consistent with our result - the bulk literature values do not lead to good fits, see also new supplemental Figs. S1-S4. Still, as a useful orientation, we included a sketch of the spectral dependence of the refractive index (data taken from reflectivity measurements of liquid bulk helium, Surko et al., Phys. Rev. Lett. 23, 842–846 (1969); Lucas et al., Phys. Rev. B 28, 2485–2496 (1983) and tabulated values from NIST database), cf. new Fig. 3d.

Concerning the interpretation of the optimized refractive indices we would like to recall that in our optimizations two “good solutions” are found with comparably small mean square displacement (see discussion above (2a) and Figs. R3-R5). We included a discussion of this issue in the main manuscript and the supplemental material. In short, the reason for the existence of two solutions is the close proximity of the 13th and 15th harmonic and their similar intensity, in addition to our experimental uncertainties. The roles of the 13th and 15th harmonic can be interchanged in the optimization process to some extent – an ambiguity that cannot be resolved with the current data but can be overcome in refined future experiments.

changes to the manuscript:

- To indicate the vicinity of the harmonics to the He1s2p resonance, we added a sketch of the refractive indices of bulk liquid helium to Fig. 3d of the revised manuscript, based on the combined available literature values.
- The issue of the remaining ambiguity (2 possible solutions) is implemented; see changes discussed in the response to the second referee (2a).

(3b) Among the three types of images, what type of diffraction accounts for the straight lines? A simulation of the rod shaped droplet lined up with the diffraction plane should be provided for reference, either in the main text or supplementary material. A statistics of the images for in-plane and out-of-plane rods and spherical droplets should be provided.

We thank the referee for requesting a discussion of the straight streaks, as the resulting analysis helped to make the assignment of the patterns to pill shaped structures even more obvious and more easily accessible.

Figure R11 3D simulation of an oblate wheel and a prolate pill which are not tilted out of the scattering plane. Even at 0°, a clear difference is visible, which results from the tomographic nature of wide angle scattering. Basically, the rod-type structure looks the same from the upper/lower edge of the detector compared to from the center.

In fact, both, the observed straight as well as the bent streak patterns in our data can be assigned to prolate structures (cf. also answer to second referee (2f)). As can be seen in Fig. R11, straight lines are created for 0° tilt angle of both wheel and pill-shaped structures.

However, the tomographic nature of wide angle scattering reveals clear differences between the two cases, as the streaks decay much faster for a wheel than for a pill-shape.

Moreover, the statistics of straight vs. bent streaks supports our assignment. About 90% of all observed streaked patterns (68 patterns with streaks, 7 straight ones) exhibit a bending, with varying curvature. The simulations in Fig. R12 show that tilt angles between ca. 70° to 20° create clear bent streaks in the diffraction patterns. Below 10° tilt angle, the streaks would appear straight. Tilt angles above 80° do not create streaks and are thus not contained in the above selected patterns.

As a result of these considerations, one would expect that up to 1/8 (0-10° out of 0-80°) of the patterns display straight streaks and 7/8 bent streaks when assuming rod-like structures with random orientation, in good agreement with our data.

Figure R12 Simulated single-color wide-angle diffraction patterns for different tilt angles of a prolate pill-shaped structure.

changes to the manuscript:

- We added the information from Fig. R11 to Fig. 4 of the revised manuscript.
- In the manuscript we state accordingly *“Although less obvious, also the analysis of observed straight streak patterns indicates that they can only be explained by prolate structures, as the observed streak signal is visible until the edge of the detector (cf. for example Fig. 2e). Figs. 4e and f show a comparison of diffraction patterns for pill and wheel shaped structures that are aligned to the scattering plane. The tomographic nature of wide angle scattering reveals that the streaks decay much faster towards larger scattering angles for a wheel than for a pill-shaped particle.”*
- Fig. R12 and the associated discussion were added to the supplement, section 3.
- In the manuscript we correspondingly reference *“we find in the majority of our wide-angle scattering patterns a pronounced crescent-shaped bending of the streaks (statistics see Fig. 2 and supplemental section 3)”*

(3c) Pg. 2, line 33, I am not sure if the imaging method in this article can be considered “tomographic” since no cross-sectional information is obtained.

While we implemented essentially all of the other suggestions of the referee, we would like to keep the expression “*tomographic*” due to the following reason: We agree that the measured diffraction pattern does not provide a cross-sectional view directly. This, however, is also the case in computer tomography (CT). Access to the cross-sectional information in CTs is obtained in the reconstruction process of the data recorded for multiple projections. In our case, contributions from multiple projections are contained in one scattering image [Barke2015], allowing the reconstruction of at least limited cross-sectional information, i.e. the 3D surface. In fact, tomographic techniques are often used to reconstruct the 3D shape of objects. In this sense we prefer to keep the term “*tomographic*”.

(3d) Pg. 2, line 38, unclear sentence ‘would allow to combine...’

We changed that sentence to... *“Using XUV and soft X-ray high harmonic generation (HHG) sources for single-shot nanoparticle CDI holds the promise to combine the nanoscale structural imaging capabilities of CDI with the exquisite temporal, spectral, and phase control inherent in the use of optical lasers...”*

(3e) Pg. 3, sentence start with an abbreviation.

We changed that sentence to *“The brightness of HHG sources is typically...”*

(3f) Pg. 4, line 64, ‘A dataset of 2300 ...’, should state the total images taken, the hit rate, and the total number of bright images.

We shifted the requested statistics information from the methods section to the main text. The sentence has been changed to: *“Within 3×10^5 single-shot measurements, 2300 bright patterns with distinct structures were obtained and another 12700 recorded images contained weak, unstructured scattering signal. A selection of exemplary diffraction patterns is displayed in Fig. 2.”*

(3g) Pg. 5, line 96, ‘using higher energy and/or lower wavelength drivers’, what do the authors mean by ‘drivers’? Lasers?

The word was substituted by *“lasers to drive the HHG process”*.

(3h) Pg. 5, line 100, ‘these three main types of patterns’: state the three main types explicitly; ‘rings, elliptical and streak patterns,’.

We implemented this suggestion.

(3i) Pg. 8, line 150, starting a sentence with ‘30%’?

We changed the beginning of this sentence to *“A fraction of 30%...”*

(3j) Pg. 9, line 184, “230 mm away from the mirror”: does it mean that the focal length is 230 mm?

Indeed, the focal length is 230 mm. We changed the sentence from *“The first of the latter two toroidal mirrors (radii 2650 mm x 79 mm) focuses the XUV beam approximately 230 mm away from the mirror whereas the last toroidal mirror (radii 3620 mm x 109.2 mm) is placed 680 mm away from the focus and is used to relay image the focus of the first toroidal mirror into the experimental chamber at a distance of 585 mm.”*

to *“The first of the latter two toroidal mirrors (radii 2650 mm x 79 mm) has a focal length of 230 mm. Subsequently the last toroidal mirror (radii 3620 mm x 109.2 mm) is placed 680 mm away from the focus to relay image the focus of the first toroidal mirror into the experimental chamber at a distance of 585 mm.”*

(3k) Pg. 10, line 204, ‘statistics indicate’: what statistics?

We extended the sentence to provide an explicit statement: *“Within 3×10^5 single-shot measurements, 2300 bright patterns with distinct structures were obtained and another 12700 recorded images contained weak, unstructured scattering signal. These statistics indicate that the experiment is performed in the single-particle limit as the probability to have two droplets in the focus at the same time is lower than 2 ‰.”*

(3l) Pg. 11, line 208 – 209, why do the authors assume the detected intensity to scale in that formula? Why use $\alpha = 0.5$?

When used in femtosecond CDI measurements [Bostedt2012], MCP-phosphor-detectors show a nonlinear detection efficiency that has to be corrected for quantitative analysis [Barke2015]. The exponent of the exponential correction function in our analysis was fitted such that the corrected patterns from spherical droplets show the well-known q^{-4} dependence as expected from Porod’s law [Porod1951, Sorensen2000]. This procedure led to the mentioned value of $\alpha = 0.5$. We agree that the motivation was too compressed and explain the idea better in the revised version.

changes to the manuscript:

- In the manuscript we now explain *“In addition, the measured data must be corrected for the nonlinear detection efficiency of the MCP [Bostedt2012, Barke2015]. Previous work*

has shown that the saturation effect can be described by an exponential efficiency function [Barke2015] such that the detected signal intensity, I_{det} , is connected to the true experimental intensity, I_{exp} , via $I_{det}=I_{exp}^\alpha$. The nonlinearity exponent $\alpha=0.5$ has been found by matching the angular decay of the envelope of scattering profiles from spherical droplets to the universal q^{-4} decay behavior predicted by Porod's law [Porod1951, Sorensen2000]."

(3m) *In Fig. 1, a problem with the labeling for 'IR-blocker, MoSi mirror...'*

The label was exchanged for *"IR blocked by a Mo/Si mirror and 100 nm Al"*

(3n) *In general, I find the figures are difficult to visualize on my computer, and impossible to print, particularly Fig. 2. Too many not quite relevant pictures are placed together without much logic.*

All figures were revised according to the proposed changes. We did our best to improve the image quality to maximum clarity. The revised versions should be easily accessible and printable.

changes to the manuscript:

- In Fig. 1 we changed one label (see also (3m)) and uploaded the graphic with a high resolution.
- An overview over the different types of diffraction images is now given in the new Fig. 2
- Fig. 3, old Fig. 2, has been reduced to one relevant example of a pattern with corresponding profile and fit solutions in addition to the measured XUV spectrum and the requested helium bulk refractive indices.
- Fig. 4, old Fig. 3, has also been substantially revised, now giving a more intuitive approach to the origin of bent streak patterns and showing the proposed comparison of straight streaks from pills and wheels.

Finally we would like to thank again all referees for the comments, suggestions, and constructive criticism. We feel that our efforts for the revision have improved our manuscript substantially and we hope that it will now be eligible for publication in Nature Communications.

Reviewers' comments:

Reviewer #2 (Remarks to the Author):

The authors have made a deliberate effort to revise the ambiguities raised in my earlier comments. It is unfortunate that a closer look at the data produced (at least) two potential classes of refractive indices for superfluid helium. Nevertheless, the fact that the authors could collect substantial data to identify these classes using CDI from an HHG source is a remarkable testament to the technique's capability.

With regards to the manuscript's primary message that HHG-based CDI can provide evidence to help resolve similar ambiguities in scattering factors, this revised manuscript is more than satisfactory. Further, the manuscript's comprehensive description of the hurdles that need to be overcome before such resolution is achieved is commendably forthcoming.

Reviewer #3 (Remarks to the Author):

The authors addressed most of the problems from the previous review. There are still a few confusing points about the manuscript, mostly about the ambiguities in the figures.

1. Fig. 1: use arrows to point to the three elements of the "coma-corrected ...", since without reading carefully in the section "method", it is unclear what are the two gray rectangular prisms in the figure, and if the golden one is the aforementioned mirror. In addition, the caption states "After removing ...", while the IR block is in-between the mirror system.
2. Fig. 3 and the resulting refractive indices: what are the values of the resulting indices? The text stated that there were two possible solutions to the fitting, but do both fitting results, i.e., the values of the refractive indices, physically make sense? Moreover, the dispersion properties of helium have already been calculated, and the fitting results should be compared with the theoretical value, and possible reasons for the difference should be offered.
3. Fig. 4: overlay the pictures with the coordinate system: which direction is the "optical axis", which is the diffraction plane, and which is the tilt angle? Part "c" is very confusing: the green and blue represent the two wavelength components, and should they both go in the forward and sideways directions?
4. Pg. 5, line 87, the fitting parameters include (iv) the intensity of the XUV – should this be just a scaling factor?

----- **Response to referee 2** -----

Referee 2: The authors have made a deliberate effort to revise the ambiguities raised in my earlier comments. It is unfortunate that a closer look at the data produced (at least) two potential classes of refractive indices for superfluid helium. Nevertheless, the fact that the authors could collect substantial data to identify these classes using CDI from an HHG source is a remarkable testament to the technique's capability.

With regards to the manuscript's primary message that HHG-based CDI can provide evidence to help resolve similar ambiguities in scattering factors, this revised manuscript is more than satisfactory. Further, the manuscript's comprehensive description of the hurdles that need to be overcome before such resolution is achieved is commendably forthcoming.

We are very happy that we were able to resolve the referee's concerns and thank the referee for her or his very supportive evaluation.

----- **Response to referee 3** -----

(Comments from Referee 3 are printed in **green** and our response in **black**)

Referee 3: The authors addressed most of the problems from the previous review. There are still a few confusing points about the manuscript, mostly about the ambiguities in the figures.

We thank the referee for the thorough evaluation of our revised manuscript and the well-considered suggestions to enhance the comprehensibility and to avoid ambiguities.

(3a) Fig. 1: use arrows to point to the three elements of the “coma-corrected ...”, since without reading carefully in the section “method”, it is unclear what are the two gray rectangular prisms in the figure, and if the golden one is the aforementioned mirror. In addition, the caption states “After removing ...”, while the IR block is in-between the mirror system.

Figure R1 Revised setup figure with clarified positions of IR-removing and focusing components.

We followed this suggestion and changed the figure and the caption accordingly such that all elements are now explicitly labeled.

changes to the manuscript:

- We have revised Fig. 1 as shown in Fig. R1.
- The caption now states “The copropagating NIR is removed via a Mo/Si mirror and a thin aluminum filter. The beam is focused to a small spot ($w_0 = 10 \mu\text{m}$) using a coma-correcting system of three gold-coated toroidal mirrors [Frassetto2014].”

(3b) Fig. 3 and the resulting refractive indices: what are the values of the resulting indices?

The fit presented in Fig. 3b (with 13th harmonic dominant) results in a radius of 380 nm and a refractive index at 20.4 eV of $n_{13} = 0.9256 + i0.0179$ and a refractive index at 23.5 eV of $n_{15} = 1.3979 + i0.0433$. The fit in Fig. 3c (with the 15th harmonic dominant) results in a radius of 445 nm and a refractive index at 20.4 eV of $n_{13} = 1.1899 + i0.04204$ and a refractive index at 23.5 eV of $n_{15} = 0.9426 + i0.0309$. All fitting solutions for r , n_{13} and n_{15} are given in the supplement; see Supplemental Figures 4-7. In Supplemental Figure 4, also the literature values for the refractive index are shown for comparison. Please note that in the Methods section, the average refractive indices for the solutions where the 13th harmonic is dominant (solution with smaller

residuals) are explicitly given, because these values are used for a subsequent calculation, i.e. the simulations of nonspherical droplets in Figure 4 of the main manuscript.

changes to the manuscript:

- We have added the following statement to the caption of Figure 2 “The resulting refractive indices of these and all other fits are given in Supplemental Figure 4.”
- We have explicitly marked the results displayed in Figure 2 of the main manuscript as squares in Supplemental Figure 4. We have added this information to the figure caption, stating “The solutions for the fits presented in Fig. 2 of the main manuscript are indicated by squares”.

(3c) The text stated that there were two possible solutions to the fitting, but do both fitting results, i.e., the values of the refractive indices, physically make sense?

The solutions with dominant 13th harmonic give slightly better fits with smaller residuals in the majority of cases. On the other hand, the refractive indices of the solutions with dominant 15th harmonic lie closer to the literature values. However, based on our current experiment with two harmonics close to the resonance and considering the limited signal-to-noise ratio of our data we cannot fully exclude one of the two solutions. We added these two points, which were in the revised version only stated in the supplemental material, to the main text.

changes to the manuscript:

- The paragraph in the main manuscript now reads “We note that while the residuals of the fits are slightly smaller for the solution with dominant 13th harmonic, the refractive indices for the solution with dominant 15th harmonic lie closer to the literature values. However, we cannot fully exclude one of the two solutions. In order to resolve such ambiguity in the optimization, future systematic studies are required with only one of the strong harmonics being near-resonance and/or with substantially better signal to noise ratio.”

(3d) Moreover, the dispersion properties of helium have already been calculated, and the fitting results should be compared with the theoretical value, and possible reasons for the difference should be offered.

The combined available literature values are sketched in Fig. 3e. The calculated dispersion properties from the databases NIST (<http://physics.nist.gov/PhysRefData/FFast/html/form.html>) and Henke (http://henke.lbl.gov/optical_constants/getdb2.html) are known to be not reliable at small photon energies and in particular close to resonances (this is explicitly stated by the authors of those databases, see http://henke.lbl.gov/optical_constants/intro.html)

and <http://physics.nist.gov/PhysRefData/FFast/html/form.html>). Therefore the calculated values from those databases were used only for the regions far from the resonance, i.e. at the 11th and 17th harmonic. These values are also listed in the Methods section, because they were used as input parameters for the fitting routine.

In the vicinity of the resonance the reflectivity of liquid helium has been measured by Surko and coworkers (Surko, C. M., Dick, G. J., Reif, F. & Walker, W. C. Spectroscopic study of liquid helium in the vacuum ultraviolet. *Phys. Rev. Lett.* 23, 842–846 (1969)) and the refractive indices have been calculated based on this measurement by Lukas and coworkers (Lucas, A. A., Vigneron, J. P., Donnelly, S. E. & Rife, J. C. Theoretical interpretation of the vacuum ultraviolet reflectance of liquid helium and of the absorption spectra of helium microbubbles in aluminum. *Phys. Rev. B* 28, 2485–2496 (1983)). The data have been used for the center part of Fig. 3e. The explicit values at the 13th and 15th harmonic as extracted from Lukas et al. are given in Supplemental Figure 4 together with all fitted refractive indices. Please note that in general deviations of the refractive indices of droplets from bulk data are expected to occur due to finite size effects (Joppien, M., Karnbach, R. & Möller, T. Electronic excitations in liquid helium: The evolution from small clusters to large droplets. *Phys. Rev. Lett.* 71, 2654–2657 (1993)) that may shift and broaden the resonances.

We feel that the presentation of a summary of available literature values for bulk helium in Fig. 3e and the statement “While the optical properties of bulk liquid helium have been measured and calculated close to the 1s-2p transition of helium 32, 33 (see Fig. 3e), the dielectric function of the nanodroplets is completely unknown and expected to vary substantially with droplet size [Joppien].” give appropriate credit to the existing work on bulk Helium and provides a useful reference motivating the size dependence. We feel that together with the following statement “In fact, we find that fits using the bulk literature values for the refractive indices at the corresponding harmonic wavelengths cannot reproduce the observed diffraction patterns (see Supplemental Note 1 and Supplemental Figure 1).”, sufficient context is provided to understand the fundamental problem and the probable reason, i.e. deviations due to the size dependent shifts of the relevant resonances/bands in the droplets. We would like to refrain from further detailed discussion of possible implications of our results regarding the size-dependent evolution because we feel that due to the ambiguity in the fitting results, this would be too speculative.

(3e) Fig. 4: overlay the pictures with the coordinate system: which direction is the “optical axis”, which is the diffraction plane, and which is the tilt angle?

The optical axis of the XUV beam is directed into image plane for all simulated geometries, i.e. Fig. 4 b, d, e and f. The tilt angle is defined as the angle between the symmetry axis of the

respective particle, i.e. the semi-major axis of pill-shaped particles and the semi-minor axis of wheel-shaped particles.

We appreciate the idea to clearly specify the used coordinate system and the definition of the tilt angle. To avoid any ambiguity, we decided to explicitly state this information in the figure caption.

changes to the manuscript:

- The caption now reads “**Unique identification of prolate, pill-shaped structures.** (a) Measured image and (b) matching simulation result of the wide-angle diffraction of a pill-shaped prolate droplet, that is visualized in yellow. The optical axis of the XUV beam is directed into the image plane, the tilt angle between the symmetry axis of the particle and the optical axis is 35°; the semi-minor axes were set to $a = b = 370$ nm and the semi-major axis $c = 950$ nm; for optical parameters see Methods. (...) (d) Simulated wide-angle diffraction image of a wheel-shaped oblate particle. If the particle’s symmetry axis is neither oriented along the optical axis nor perpendicular to it, the diffraction patterns exhibit straight streaks to only one side (parameters: semi-major/-minor axes as in (b), tilt angle between the symmetry axis and the optical axis (directed into the image plane) 80°). (e,f) Comparison of simulated wide-angle diffraction images of a prolate (e) and an oblate structure (f) aligned to the scattering plane, i.e. at 90° tilt angle between the symmetry axis and the optical axis, other parameters as in (b),(d). Though the 2D projections are similar and the 2D outlines identical, the intensity distributions of the straight streaks are clearly different and decay much faster for “wheels” than for “pills”.”

(3f) Part “c” is very confusing: the green and blue represent the two wavelength components, and should they both go in the forward and sideway directions?

As noted before in the caption, the different ray colors were chosen for visibility, they do not refer to wavelengths. In order to avoid confusion, we have added an additional explanation for the visualization and now present the information on the colors more prominently.

changes to the manuscript:

- The caption now reads “(c) Illustration of the origin of bent streaks occurring when a tilted rod-type structure diffracts the light. The constructive interference is analogous to the specular reflection at the surface of a macroscopic rod. Two particular bundles of constructively interfering rays are explicitly sketched, please note that the different ray colors do not refer to wavelengths, but are applied to facilitate distinction.

(3g) Pg. 5, line 87, the fitting parameters include (iv) the intensity of the XUV – should this be just a scaling factor?

The assumption of the referee is correct; the total incoming intensity is a linear scaling factor in the scattering process. To make this point clear we slightly changed the sentence.

changes to the manuscript:

- The sentence now reads “(iv) a scaling factor for the total XUV intensity”

Finally we would like to thank again Referee 3 for the constructive criticism. We feel that our efforts for the revision have improved our manuscript and particularly the figures substantially and we hope that it will now be suitable for publication in Nature Communications.

REVIEWERS' COMMENTS:

Reviewer #3 (Remarks to the Author):

The authors have addressed all of my concerns. Publish as is.